# A Study on Adolescent Smoking Prevention and Cessation Policies: Based on the Propensity Score Matching–Difference-in-Differences Method

**DOI:** 10.3390/healthcare13010030

**Published:** 2024-12-27

**Authors:** Seokmin Ji, Byungchan Moon, Younggyu Kwon, Kyumin Kim

**Affiliations:** 1Department of Health Policy & Management, College of Public Health Science, Korea University, Seoul 02841, Republic of Korea; seoky9403@korea.ac.kr (S.J.); gamgz1@naver.com (B.M.); kgkg1025@cau.ac.kr (Y.K.); 2Global Procurment Development Institute, Gyeonggi 10403, Republic of Korea; 3Center for Chung Ang Medical Education Resource Allocation, College of Medicine, Chung Ang University, Seoul 06974, Republic of Korea; 4Department of Health Administration, Gyeonggi University of Science and Technology, Gyeonggi 15073, Republic of Korea

**Keywords:** adolescent smoking, secondhand smoke, smoking advertising, propensity score matching, difference in differences

## Abstract

**Introduction**: Adolescent smoking can lead to various health problems including atherosclerosis and cardiovascular disease, making it more difficult to quit smoking during adulthood. This study aims to evaluate the effect of the ordinance by assessing adolescents’ smoking cessation behaviors and environmental conditions, according to the “Ordinance for the Prevention of Child and Adolescent Smoking and Support for Smoking Cessation”, which was enacted in 2019 in Gwangju City, South Korea, for the first time in the country. **Methods**: The data for the analysis were obtained from the 2018 and 2021 Korea Youth Risk Behavior Survey. Propensity score matching (PSM) ensured homogeneity between ordinance-adopted and non-adopted areas, followed by a difference-in-differences (DID) analysis to assess changes in adolescent smoking behavior, secondhand smoke exposure, and related advertisements. **Results**: The difference-in-differences analysis of the homogeneous treatment and control groups in 2018 and 2021 showed statistically significant reductions in the variables “experience of secondhand smoke indoors at school” in the secondhand smoke category, and “exposure to smoking advertisements in convenience stores” in the smoking advertising category. However, there were no statistically significant changes in the other eight variables, such as smoking behavior and smoking cessation advertising. **Conclusions**: The results of this study suggest that the ordinance enacted in Gwangju Metropolitan City in 2019 created a positive environment for smoking prevention in convenience stores and schools for adolescents. Furthermore, if the ordinance is supplemented with the management of smoking behaviors and smoking cessation advertisements, it is expected to achieve the main purpose of the policy, which is to protect adolescents from smoking.

## 1. Introduction

From 1988 to 1997, the smoking rate among South Korean adolescents increased from 23.0% to 35.3%. However, this trend experienced a marked decline, dropping to 18.1% in 2008, nearly half the 1997 rate, and continued to decline until 2022, reaching 6.2% for boys and 2.7% for girls [1,2]. This decline in smoking rates appears to be the positive result of comprehensive smoking cessation policies in South Korea, including the enactment of the National Health Promotion Act, nationwide smoking cessation clinics, increased cigarette taxes, and the introduction of pictorial health warnings on packaging [3]. Despite these reductions, the 2022 average adolescent smoking rate of 4.5% indicates that more than four in every 100 South Korean adolescents still smoke. It should be noted that the age of the first smoking experience and that of the onset of continuous smoking among adolescents slightly shifted from 13.2 and 14.1 years in 2019 to 13.5 and 14.2 years in 2021, respectively; however, they still show a pattern of early initiation [4,5]. Additionally, recent data from the Korea Disease Control and Prevention Agency’s Youth Risk Behavior Survey reveal an increase in the prevalence of adolescent smoking—defined as smoking cigarettes on one or more days in the past 30 days—of 0.5% from 6.5% in 2019 to 7.0% in 2022. Additionally, the prevalence of using tobacco products, including cigarettes and e-liquids, on one or more days in the past 30 days increased from 7.1% in 2019 to 8.0% in 2022 [4].

Adolescent smoking is an important determinant of overall health in adulthood [6]. Adolescent smokers report more respiratory symptoms, such as cough and phlegm, than those who do not smoke and are more likely to develop chronic health problems during adulthood, including atherosclerosis and cardiovascular diseases [7,8]. Nicotine dependence during adolescence increases the likelihood of sustained and addictive smoking behavior, complicating cessation efforts in adulthood [9]. Furthermore, adolescents who begin smoking before the age of 15 years are twice as likely to develop drug addictions and engage in criminal activities compared to those who start later [10,11]. Given the potential for adolescent smoking habits to persist into adulthood, adolescent smoking can significantly impact future societal costs [12]. In the USA, for instance, 6–18% of healthcare costs are attributable to smoking, varying by state, with smoking’s economic burden estimated at 1% of the GDP and total annual productivity losses owing to indirect costs at approximately USD 151 billion. In the United Kingdom, direct smoking-related costs are estimated to be GBP 2.7–5.2 billion, accounting for approximately 5% of the total NHS budget [12]. In response to the socioeconomic and health implications of adolescent smoking, various tobacco control policies have been implemented worldwide [13,14]. Current tobacco control policies that focus on smoking prevention and cessation are broadly categorized into price and non-price policies [15]. Price policies typically include cigarette pricing and taxes, whereas non-price strategies include creating environments conducive to smoking prevention, providing smoking cessation education, regulating smoking advertisements, and providing free smoking cessation treatments. Following Korea’s enactment of the National Health Promotion Act in the 1990s and the Framework Convention on Tobacco Control (FCTC) in 2005, a comprehensive set of tobacco control measures, incorporating both pricing and non-pricing strategies, was put into effect by the government [16,17].

To date, most studies have focused on pricing policies on reducing cigarette consumption [18,19,20]. Although increasing cigarette prices is recognized as an effective strategy for reducing consumption, relatively less attention has been given to the effectiveness of non-price policies [21]. Additionally, previous studies on tobacco control policies have largely focused on identifying factors that contribute to continued smoking habits and cessation, with only limited studies exploring the effectiveness of these policies within specific geographic contexts [22,23]. Considering that non-price policies are intended to foster environments that support smoking reduction, municipal ordinances, which essentially bind an unspecified number of people, can play an important role toward the realization of this goal [24]. While most ordinances are binding to all residents and institutions within a municipality, as well as to supervisory authorities and courts, some, such as local laws, are only effective in a certain area or self-governing in nature, subject to the will of the municipality and within their own deadlines [25]. Hence, while South Korea’s National Health Promotion Act and its enforcement regulations provide a framework for tobacco control, they may not be sufficient to coordinate and regulate all details specific to each region. In contrast, the Tobacco Control Ordinance, which can be enacted and amended by local governments, allows for a more individual, targeted, and efficient regulation of smoking within each autonomous region.

In 2019, Gwangju City enacted its first ordinance to prevent smoking among children and adolescents. This study aimed to examine the smoking cessation behaviors and environmental conditions of Gwangju adolescents’ post-ordinance to demonstrate the importance of non-price policies and provide evidence for policy development.

## 2. Methods

### 2.1. Study Design and Data Preparation

This study was conducted to evaluate the effect of the introduction of the ordinance using raw data from the 2018 and 2021 Korea Youth Risk Behavior Survey (KYRBS), and to further examine the smoking behavior of adolescents following the introduction of the ordinance; Figure 1 shows the specific study model. Propensity score matching (PSM) was performed to ensure homogeneity between the groups in the ordinance-adopted and non-ordinance-adopted areas. Next, to evaluate the effect of the ordinance on the group of localities, we used a difference-in-differences (DID) analysis to evaluate changes in smoking behavior, secondhand smoke, smoking cessation advertisements, and smoking advertisements among adolescents.

### 2.2. Data Collection

The KYRBS used in this study has been conducted annually since 2005, based on the National Health Promotion Act, and is representative of health behaviors such as diet, physical activity, smoking, and drinking among Korean adolescents. The sampling was performed through population stratification, sample allocation, and tabulation. The survey method was an anonymous self-completion online survey method, and the survey was conducted by assigning one computer to each student in the sample class in a computer room with Internet access, and randomly arranging the seats. The 2018 survey used in this study surveyed 62,823 students from 400 middle and 400 high schools, totaling 800 schools; 60,040 students from 800 schools participated in the survey, with a 95.6% participation rate based on the number of students. The 2021 survey was conducted in 800 schools, 400 middle schools, and 400 high schools; 54,848 students from 796 schools participated in the survey, representing a participation rate of 92.9% based on the number of students. The study population used in the analysis was limited to adolescents who answered “yes” to the question “Have you ever smoked a cigarette, even one or two puffs?” when asked about their smoking history. This study was approved by the Korea University Institutional Review Board (approval number: KUIRB-2024-0038).

### 2.3. Definition of Variables

Table 1 shows the specific definitions of the dependent variables, group variables, time variables, and covariates used to evaluate the impact of the ordinance. The dependent variable of the DID was divided into four categories: smoking behavior, secondhand smoke, smoking cessation advertisements, and smoking advertisements. In the smoking behavior category, variables were used for the frequency of smoking, smoking cessation attempts, and the completion of smoking cessation education. In the secondhand smoke category, variables were used for experiences with secondhand smoke at home, school, and places other than home and school. The smoking cessation advertising category used variables for the experience with smoking cessation advertisements on TV, radio, and the Internet. Regarding smoking advertisements, the variables were used for experiences with smoking advertisements in magazines, convenience stores, and supermarkets. The group variable was set as the treatment group of adolescents in Gwangju City, which was categorized into areas where the ordinance was implemented, and the control group of adolescents outside Gwangju City, which was categorized into areas where the ordinance was not implemented. As time variables, 2018, the year before the ordinance was implemented in 2019, and 2021, two years after it was implemented, were used. The covariates for propensity score matching included sex, grade, academic performance, family socioeconomic status, residential status, and mothers’ and fathers’ educational levels.

### 2.4. Statistical Analysis

DID methods are commonly used to evaluate the net impact of policy intervention [26]. Unbiased effect estimates can be obtained by comparing a group affected by a policy intervention (the treatment group) with a group unaffected by the policy intervention (the control group), and analyzing the changes over time. The DID method requires the common tendency assumption to be satisfied, which typically requires that the difference between the two groups be sufficiently small. In this study, a model that combines the PSM by Rosenbaum and Rubin (1983) with the DID method was utilized to address errors that may arise owing to selection bias [27,28]. SAS version 9.0 was used for data handling, and STATA 17 was used to analyze the DID and PSM.

#### 2.4.1. Propensity Score Matching

PSM methods have been widely used to estimate policy effects since Rosenbaum and Rubin [27]. For propensity score matching, general characteristics such as sex, grades, academic performance, family socioeconomic status, residence status, and mother’s and father’s education levels were used as control variables. The specific analysis process is as follows. First, to estimate the propensity score, a probit regression analysis was performed with the binary variables of the treatment and control groups, indicating whether the treatment was applied as the dependent variable. Subsequently, based on the estimated propensity scores, near-neighbor matching (NN matching) and Caliper Matching were applied to match individual samples with similar scores. NN matching is an appropriate method when every experimental group can be matched with at least one other group, so that fewer data points are dropped, and the number of samples in the comparison group is sufficiently larger than that in the experimental group. Lastly, to assess the matching quality, we conducted a balancing test to verify the balance of covariates between the treatment and control groups. This was performed by calculating the standardized difference between the two groups after matching. Generally, if the difference is less than 10% (0.1), the difference between the two groups is considered negligible [29].

#### 2.4.2. Difference in Differences

DID estimates the policy effect by finding pre-implementation and post-implementation changes in the experimental and comparison groups, and then subtracting the pre- change and post-change in the comparison group from the pre-change and post-change in the experimental group [30]. This study examines the policy effects using DID regression to estimate the net effect of introducing the ordinance. In this study, Gwangju City was considered the treatment group and the rest of the country the control group, to analyze how the introduction of an ordinance to prevent smoking in Gwangju affected adolescent smoking behavior in the region.

The basic DID analysis uses a regression model such as Equation (1), with a dummy variable $\left(X_{t}\right)$ for whether the ordinance was adopted (1 for treatment, 0 for control), a dummy variable $\left(X_{p}\right)$ for the observation period (0 for time before adoption, 1 for time after adoption), and the product of the two $\left(X_{t}X_{p}\right)$ as the independent variable.
(1)$$Y=\beta_{0}+{\beta_{t}X}_{t}+{\beta_{p}X}_{p}+{\beta_{tp}X_{t}X}_{p}+\in$$
where Y is the dependent variable representing the effect of the policy; $\beta_{0}$ is the constant term; $\beta_{t}$, $\beta_{p}$, and $\beta_{tp}$ are the regression coefficients of the independent variables; and ∈ is the error term. When analyzing using this basic model, $\beta_{tp}$ is the estimate of the effectiveness of the policy.

## 3. Results

### 3.1. General Characteristics of the Study Population

Table 2 and Table 3 present the general characteristics of the treatment and control groups before and after PSM in 2018 and 2021. In 2018, the numbers of cases in the treatment and control groups before PSM were 183 and 3539, respectively, and the number of cases in the treatment and control groups after PSM were 183. In 2021, the number of cases in the treatment and control groups before PSM was 138 and 2266, respectively, and the number of cases in the treatment and control groups after PSM was 138.

The standardized mean difference values of the variables after propensity score matching according to the Chi-square test show that there were no statistically significant differences in most variables including sex, academic performance, family socioeconomic status, residence status, and father’s and mother’s education levels. This indicates that the demographic characteristics of the treatment and control groups are similar.

The general characteristics common to the treatment and control groups in 2018 and 2021 included more males than females, more high school seniors than juniors, and more students in the middle and lower academic grades. The family’s socioeconomic status was more likely to be in the middle class, and the most common residential status was living with family. Mothers’ and fathers’ education levels differed between 2018 and 2021, with the highest number of students in 2018 having both mothers and fathers with a college degree or higher, whereas the highest number of students in 2021 did not respond to questions about mothers’ and fathers’ education.

Figure 2 and Figure 3 show the probability distributions of the kernel density functions of the treatment and control groups before and after PSM in 2018 and 2021. In 2018, the probability distribution of the kernel density function showed irregular differences in the range of 0.04 to 0.07 before matching; however, after matching, it was almost identical to the treatment group (Figure 2). Additionally, the probability distribution of the kernel density function in 2021 showed irregular differences in the range of 0.05 to 0.09 before matching, but almost matched the treatment group after matching (Figure 3). Therefore, it can be concluded that PSM was used to construct a control group of smokers outside Gwangju City with demographic characteristics similar to those of the policy intervention group.

### 3.2. Results of DID Analysis for Evaluating Policy Effectiveness

Table 4 shows the DID of the treatment and control groups according to ten dependent variables. First, the interaction term evaluating the effect of the policy showed a statistically significant decrease in the treatment group compared with the control group for two variables including secondhand smoke in schools and smoking cessation advertisements in convenience stores. These variables were assessed as outcomes of policy interventions aimed at creating a smoking prevention environment and providing smoking cessation support for children and adolescents. Next, the time variable showed a statistically significant decrease as time increased for smoking cessation education, secondhand smoke at school and outside home, and smoking advertisements in magazines and supermarkets. However, there was a significant increase in smoking cessation advertisements over time.

Overall, statistically significant reductions were found in the interaction terms for secondhand smoke in schools and convenience store smoking advertisements, which were assessed as the effects of policy interventions to create a smoking prevention environment and smoking cessation support for children and adolescents.

## 4. Discussion

Aiming to evaluate the effect of Gwangju’s “Ordinance for the Prevention of Child and Adolescent Smoking and Support for Smoking Cessation” by assessing adolescents’ smoking cessation behaviors and environmental conditions, this study found the following.

First, we employed PSM and a DID analysis to determine changes in adolescent smoking behaviors and environmental conditions post-ordinance implementation. Statistically significant time-varying differences were observed in some of the measures of “experience with secondhand smoke” and “exposure to smoking advertisements” (*p* < 0.05), aligning with previous studies showing a reduction in exposure to smoking advertisements following tobacco control policies [31,32,33]. Kasza et al. (2011) reported a general decline in smoking advertisement perceptions post-enactment of tobacco controls, with the greatest decrease occurring immediately after the ban was implemented. Furthermore, awareness of smoking advertisements was generally similar across different socioeconomic groups, with a sharper decline in the high social economic status (SES) group than in the low-SES group. Usidame et al. (2019) found that retail outlets in municipalities with comprehensive local tobacco control policies were more likely to display fewer smoking advertisements and feature fewer categories of smoking advertisements. Holmes et al. (2022) compared the implications of tobacco control enactments in the Californian counties of Alameda and San Francisco, revealing substantial reductions in smoking advertising across both locations and between cities with and without comprehensive policies. This included a broad range of tobacco products such as menthol and non-menthol cigarettes, cigars, smokeless tobacco, and e-cigarettes. The most substantial reductions in both external and internal advertising occurred in cities with regulations, with relatively small reductions and some increases occurring in cities without such regulations. It has been reported that increased exposure to smoking advertisements heightens adolescents’ curiosity or desire to smoke. Moreover, the number of retail outlets near their school correlates with greater exposure to smoking advertisements and increased awareness of tobacco product brands [34]. These findings suggest that effective deterrents to retailers near schools should be explored in places where youth are more likely to be exposed to smoking advertisements.

Second, our results are also consistent with previous studies demonstrating a decline in exposure to secondhand smoke in schools following the implementation of tobacco control policies [35,36]. Seo et al. (2011) found that the adoption of a smoke-free campus policy at Indiana University (treatment group) significantly reduced the prevalence of smoking among students from 16.5% to 12.8%, in contrast to Purdue University (control group), where no such policy was in place. Moreover, there was a significant decrease in perceptions of peer tobacco use at Indiana University, unlike at Purdue, where perceptions increased. Similarly, Chuang and Huang (2012) reported significant changes in college students’ smoking behavior in Taiwan after instituting a stringent no-smoking campus policy. These changes included a reduction in smoking behaviors, shifts in social norms, increased concern for the health of others, and an increased sense of compliance with the law. These findings suggest that creating environments such as smoke-free campuses can help reduce exposure to tobacco use and establish non-smoking as an everyday social norm. This is particularly impactful given that adolescents are highly susceptible to social influences, with their behaviors and attitudes strongly influenced by their immediate environments. Implementing these policies can play an important role in preventing adolescents from smoking and encouraging individuals who already smoke to cease smoking.

Third, “home” in the “experience of secondhand smoke” category was statistically non-significant, aligning with previous studies that tobacco control laws do not necessarily apply changes in smoking behaviors to the home [37,38]. For instance, Zheng et al. (2017) found that when smoking was prohibited in most public places, parents often resorted to smoking at home, thereby increasing secondhand smoke exposure among children. Kim et al. (2023) reported no significant changes in secondhand smoke exposure or primary smoking prevalence in the home, suggesting that tobacco control policies may have difficulty penetrating private spaces, such as homes, effectively. These findings imply that public place smoking bans alone are insufficient. A comprehensive approach is essential to effectively address smoking and secondhand smoke exposure in homes. This approach should include educational and awareness campaigns, efforts to shift social attitudes toward smoking at home, and support programs to assist smokers in quitting.

Fourth, “smoking frequency” and “quitting attempts” were statistically non-significant, which contrasts with findings from previous studies. In the USA, research has demonstrated that stringent smoking control ordinances reduce smoking adoption and decrease the likelihood of progression from experimental to established smoking [39,40]. Furthermore, an analysis conducted in the UK concluded that smoking control legislation may be associated with a reduction in regular smoking among school-aged children [41]. Similarly, an Australian study identified a direct link between smoke-free policies and a decline in adolescent smoking rates between 1990 and 2015 [42]. However, Anger et al. (2010) found no significant effect of smoking control laws on smoking behavior. Anger et al. (2010) examined the effects of state-by-state public smoking bans in Germany on smoking behavior and found that such bans had no effect on overall smoking behavior [43]. These discrepancies suggest that the effectiveness of smoking prevention policies can vary across countries or regions, indicating that smoking prevention ordinances in a region may not guarantee statistical significance for a given behavioral variable. These findings suggest that it is imperative to understand the diverse factors influencing smoking behavior and to develop comprehensive tobacco control policies and programs tailored to these specific contexts. This approach ensures that the policies are not only designed to meet statutory requirements but are also effective in changing behavior at the community level.

Finally, this study has several limitations. First, the study does not establish a direct causal pathway between the reduction in adolescent smoking prevalence and the implementation of smoking prevention policies. This limitation is primarily due to methodological constraints and the scope of data available; thus, future research should aim to collect more extensive and detailed data, analyze the interactions of various variables, and conduct long-term follow-up studies to better elucidate this causal relationship. Furthermore, the analysis did not comprehensively address the specific reasons for changes in the age of smoking initiation or the persistent decline in smoking prevalence. Therefore, there is a need for further research to explore the social and psychological factors associated with adolescent smoking behavior and to assess the effectiveness of prevention strategies targeting smoking triggers. Second, this study was unable to track the impact of the ordinances over time due to the nature of the KYRBS, which is a cross-sectional survey that samples a different group each year. This limits its ability to precisely analyze the temporal impact of policies. Future research should employ long-term panel data to more accurately track changes in smoking prevalence and cessation attempts over time. This approach would enable a detailed analysis of both the short- and long-term effects of policies, significantly contributing to the evaluation and effectiveness of smoking prevention efforts. It is also necessary to utilize data disaggregated by different age groups and geographic regions to further examine whether the effects of policies are more pronounced in certain age groups or regions, or whether there are unexpected negative effects. Such an in-depth analysis can help policymakers create more effective and customized tobacco control policies. Third, there were no statistically significant findings regarding the impact of the ordinance on adolescents’ “frequency of smoking” and “smoking cessation attempts”. These findings suggest that tobacco control policies may have limited effectiveness and have important implications for the design of policies aimed at preventing adolescent smoking. Future policy designs should incorporate a broader understanding of the various social and psychological factors that influence adolescent smoking behavior, rather than relying solely on legal restrictions. Strategies could include implementing school- and community-based education programs, targeted smoking cessation campaigns tailored to adolescent populations, and initiatives that encourage active parental and community involvement.

## 5. Conclusions

This study differs from previous studies in that it used a combined PSM-DID model to evaluate the effect of non-price policies under a smoking prevention ordinance for young people in Gwangju, South Korea. By assessing changes in key health-related behaviors, such as smoking cessation and exposure to secondhand smoke, this study provides a comprehensive evaluation of the ordinance’s impact on creating a supportive environment that promotes healthier behaviors among adolescents.

Our conclusions indicate that the effectiveness of smoking prevention policies may vary significantly depending on local and cultural contexts. The successes and limitations observed in Gwangju City’s policies provide valuable insights for policymakers in other regions and countries aiming to reduce adolescent smoking rates and implement effective smoking prevention strategies. Moreover, when these ordinances are supported by stringent controls on smoking behavior and targeted smoking cessation advertisements, they are more likely to achieve their intended goal of preventing adolescent smoking.

## Figures and Tables

**Figure 1 healthcare-13-00030-f001:**
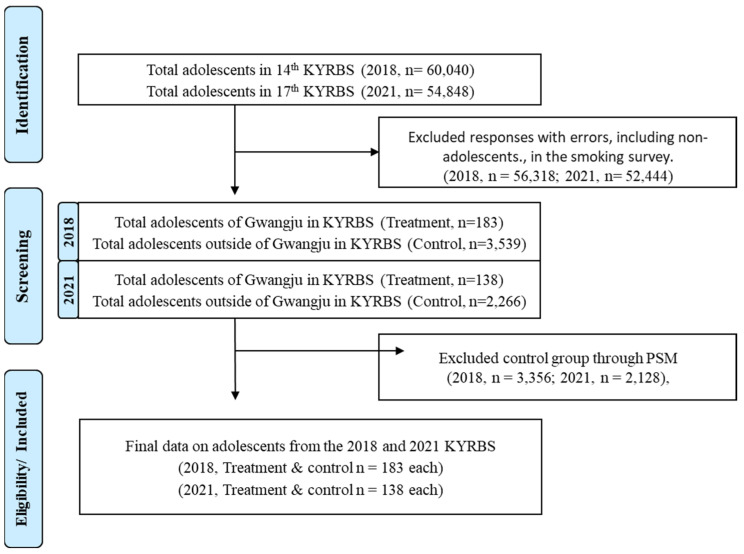
Flow diagram of participant selection.

**Figure 2 healthcare-13-00030-f002:**
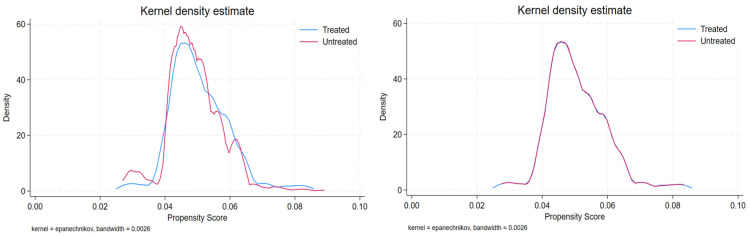
Kernel density plots of PSM, before and after (2018).

**Figure 3 healthcare-13-00030-f003:**
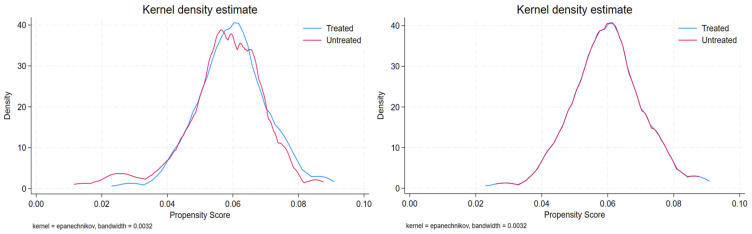
Kernel density plots of PSM, before and after (2021).

**Table 1 healthcare-13-00030-t001:** Variable description.

Analysis	Variables		Definition
Difference in differences	Smoking action	Smoking frequency	1–2 days per month
			3–5 days per month
			6–9 days per month
			10–19 days per month
			20–29 days per month
			Every day
		Cessation attempts	Yes, in the last 12 months
			No, in the last 12 months
		Cessation education attempts	Yes, in the last 12 months
			No, in the last 12 months
	Secondhand smoke	Home	1 day per week
			2 days per week
			3 days per week
			4 days per week
			5 days per week
			6 days per week
		School	7 days per week
		Other than home and school	Every day
	Smoking cessation		Yes, in the last 12 months
	advertising experience		No, in the last 12 months
	Smoking advertising experience	Magazine	Yes, in the last 30 days
		Convenience stores	No, in the last 30 days
		Supermarket	
	Group variable		Treatment, Gwangju
			Control, other than Gwangju
	Period variable		2018
			2021
Propensity score matching	Sex		Male
			Female
	Grade		1st year of middle school
			2nd year of middle school
			3rd year of middle school
			1st year of high school
			2nd year of high school
			3rd year of high school
	Academic performance		Upper
			Upper middle
			Middle
			Middle lower
			Lower
	Family socioeconomic status		Upper
			Upper middle
			Middle
			Middle lower
			Lower
	Residence status		With family
			With relative
			Boarding, rooming
			Dormitory
			Child care
	Mother and		≤Middle school
	father’s		High school
	education level		≥College
			Not sure
			No response

**Table 2 healthcare-13-00030-t002:** General characteristics before and after PSM (2018).

		Before Matching (2018)					After Matching (2018)				
		Treatment group		Control group		*p*	Treatment group		Control group		*p*
		N	%	N	%		N	%	N	%	
		183	4.9	3539	95.1		183	50.0	183	50.0	
Gender	Male	132	72.1	2514	71.0	0.101	132	72.1	130	71.0	0.817
	Female	51	27.9	1025	29.0		51	27.9	53	29.0	
Grade	Middle school1	2	1.1	74	2.1	0.239	2	1.1	11	6.0	0.000 **
	Middle school2	21	11.5	273	7.7		21	11.5	32	17.5	
	Middle school3	24	13.1	449	12.7		24	13.1	48	26.2	
	High school1	29	15.9	692	19.6		29	15.9	28	15.3	
	High school2	41	22.4	910	25.7		41	22.4	27	14.8	
	High school3	66	36.0	1141	32.2		66	36.0	37	20.2	
Academic performance	Upper	26	14.2	360	10.2	0.260	26	14.2	26	14.2	0.988
	Upper middle	28	15.3	500	14.1		28	15.3	30	16.4	
	Middle	37	20.2	789	22.3		37	20.2	33	18.0	
	Middle lower	55	30.1	994	28.1		55	30.1	56	30.6	
	Lower	37	20.2	896	25.3		37	20.2	38	20.8	
Family socioeconomic status	Upper	18	9.8	465	13.1	0.085	18	9.8	19	10.4	0.999
	Upper middle	60	32.8	845	23.9		60	32.8	60	32.8	
	Middle	70	38.3	1463	41.4		70	38.3	68	37.2	
	Middle lower	26	14.2	546	15.4		26	14.2	27	14.7	
	Lower	9	4.9	220	6.2		9	4.9	9	4.9	
Residencestatus	With family	161	88.0	3208	90.6	0.533	161	88.0	163	89.1	0.853
	With relative	4	2.2	64	1.8		4	2.2	3	1.7	
	Boarding, rooming	8	4.4	91	2.6		8	4.4	5	2.7	
	Dormitory	7	3.8	101	2.9		7	3.8	7	3.8	
	Child care	3	1.6	75	2.1		3	1.6	5	2.7	
Father’s education level	≤Middle school	6	3.3	113	3.2	0.324	6	3.3	4	2.2	0.964
	High school	53	29.0	1119	31.6		53	29.0	54	29.5	
	≥College	89	48.6	1465	41.4		89	48.6	92	50.3	
	Not sure	19	10.4	511	14.4		19	10.4	19	10.4	
	No response	16	8.7	331	9.4		16	8.7	14	7.6	
Mother’s education level	≤Middle school	3	1.6	86	2.4	0.168	3	1.6	2	1.1	0.918
	High school	57	31.2	1261	35.6		57	31.2	50	27.3	
	≥College	88	48.1	1381	39.0		88	48.1	94	51.4	
	Not sure	22	12.0	470	13.3		22	12.0	24	13.1	
	No response	13	7.1	341	9.7		13	7.1	13	7.1	

** *p* < 0.01.

**Table 3 healthcare-13-00030-t003:** General characteristics before and after PSM (2021).

		Before Matching (2021)					After Matching (2021)				
		Treatment group		Control group		*p*	Treatment group		Control group		*p*
		N	%	N	%		N	%	N	%	
		138	5.7	2266	94.3		138	50.0	138	50.0	
Gender	Male	88	63.8	1541	68.0	0.301	88	63.8	80	58.0	0.324
	Female	50	36.2	725	32.0		50	36.2	58	42.0	
Grade	Middle school1	3	2.2	80	3.5	0.876	3	2.2	2	1.5	0.841
	Middle school2	10	7.3	181	8.0		10	7.3	13	9.4	
	Middle school3	17	12.3	278	12.3		17	12.3	22	15.9	
	High school1	27	19.5	459	20.2		27	19.5	29	21.0	
	High school2	43	31.2	609	26.9		43	31.2	41	29.7	
	High school3	38	27.5	659	29.1		38	27.5	31	22.5	
Academic performance	Upper	5	3.6	189	8.3	0.408	5	3.6	4	2.9	0.997
	Upper middle	18	13.1	289	12.8		18	13.1	19	13.8	
	Middle	36	26.1	558	24.6		36	26.1	36	26.1	
	Middle lower	42	30.4	636	28.1		42	30.4	41	29.7	
	Lower	37	26.8	594	26.2		37	26.8	38	27.5	
Family socioeconomic status	Upper	12	8.7	262	11.5	0.531	12	8.7	12	8.7	0.998
	Upper middle	34	24.7	546	24.1		34	24.7	34	24.6	
	Middle	58	42.0	1022	45.1		58	42.0	59	42.7	
	Middle lower	25	18.1	321	14.2		25	18.1	23	16.7	
	Lower	9	6.5	115	5.1		9	6.5	10	7.3	
Residencestatus	With family	131	94.9	2085	92.0	0.250	131	94.9	131	94.9	0.887
	With relative	2	1.5	19	0.8		2	1.5	2	1.5	
	Boarding, rooming	4	2.9	58	2.6		4	2.9	3	2.2	
	Dormitory	0	0.0	56	2.5		0	0.0	1	0.7	
	Child care	1	0.7	48	2.1		1	0.7	1	0.7	
Father’s education level	≤Middle school	1	0.7	37	1.6	0.746	1	0.7	1	0.7	0.987
	High school	31	22.5	449	19.8		31	22.5	34	24.6	
	≥College	43	31.2	657	29.0		43	31.2	44	31.9	
	Not sure	14	10.1	223	9.9		14	10.1	12	8.7	
	No response	49	35.5	900	39.7		49	35.5	47	34.1	
Mother’s education level	≤Middle school	2	1.5	29	1.3	0.771	2	1.5	1	0.7	0.979
	High school	30	21.7	494	21.8		30	21.7	29	21.0	
	≥College	45	32.6	641	28.3		45	32.6	47	34.1	
	Not sure	13	9.4	198	8.7		13	9.4	14	10.1	
	No response	48	34.8	904	39.9		48	34.8	47	34.1	

**Table 4 healthcare-13-00030-t004:** Results of DID analysis.

	**Smoking Action**						**Secondhand Smoke**					
	**Smoking Days Per Month**		**Cessation Attempts**		**Cessation Education Attempts**		**Home**		**School**		**Home and School**	
	** *β* **	**SE**	** *β* **	**SE**	** *β* **	**SE**	** *β* **	**SE**	** *β* **	**SE**	** *β* **	**SE**
Constant	5.13 **	0.14	1.71 **	0.03	1.70 **	0.03	2.74 **	0.19	2.69 **	0.18	4.21 **	0.20
Group variable	0.07	0.20	0.04	0.05	0.09	0.05	0.04	0.27	0.73 **	0.28	0.32	0.28
Time variable	0.12	0.23	0.02	0.05	−0.17 **	0.05	−0.34	0.29	−1.20 **	0.22	−0.93 **	0.29
Interaction term	0.08	0.32	−0.07	0.07	0.07	0.07	−0.12	0.40	−0.72 *	0.33	−0.19	0.41
	**Smoking Cessation Advertising Experience**				**Smoking Advertising Experience**							
					**Magazine**			**Convenience Stores**			**Supermarket**	
	** *β* **		**SE**		** *β* **	**SE**		** *β* **	**SE**		** *β* **	**SE**
Constant	1.17 **		0.03		1.15 **	0.03		1.55 **	0.04		1.32 **	0.03
Group variable	−0.03		0.04		0.01	0.04		0.05	0.05		0.09	0.05
Time variable	0.10 *		0.05		−0.10 **	0.03		0.00	0.06		−0.10 *	0.05
Interaction term	0.05		0.07		−0.01	0.05		−0.16*	0.08		−0.09	0.07

* *p* < 0.05, ** *p* < 0.01.

## Data Availability

The data can be available for a special purpose upon request to the first author of the study.

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
