# Peer review of "A Study on Adolescent Smoking Prevention and Cessation Policies: Based on the Propensity Score Matching–Difference-in-Differences Method"

_healthcare, 2024, doi:10.3390/healthcare13010030_

Round 1

Reviewer 1 Report

Comments and Suggestions for Authors

This manuscript has merits, but there are some places requiring clarification and improvement. Below are my comments:

1.      Elaborate the definition of academic performance.

2.      Modify the titles for table 2 and table 3. They are exactly the same and it is hard for the readers to get a quick sense without reading the entire text.

3.      Consider combing some response categories with very low responses.

4.      For the categories “No response” for Father/Mother education level, were they treated as missing values in the analyses? Please explain.

5.      For section 3.2., it is unclear what new information was delivered in paragraph 2?

6.      It is unconventional to start a sentence using reference number in brackets. For example, “[37] reported no significant changes in secondhand smoke exposure…”

Reviewer 2 Report

Comments and Suggestions for Authors

1. The paper needs an abstract summarizing the study findings.

2. "However, this trend experienced a marked decline, dropping to 18.1% in 2008, nearly half the 1997 rate, and continued to decline until 2022, reaching 6.2% for boys and 2.7% for girls."

Is there any explanation for the decline in smoking rates? Were there significant national policies on smoking cessation or changes in cigarette prices that contributed to this marked decline?

3. "In 2019, Gwangju City enacted its first ordinance to prevent smoking among children and adolescents."

The authors may add more details about the Gwangju City ordinance to prevent smoking, such as what specific measures it includes and the format in which it was enacted.

4. In Figure 1, n = 56,318, and 52,444 responses were excluded due to variable errors. It would be helpful for the authors to provide examples of the types of variable errors that led to the exclusion of these responses, as the large number of exclusions have reduced the final eligible sample size.
